# Ultrasound Ultrafast Power Doppler Imaging with High Signal-to-Noise Ratio by Temporal Multiply-and-Sum (TMAS) Autocorrelation

**DOI:** 10.3390/s22218349

**Published:** 2022-10-31

**Authors:** Che-Chou Shen, Feng-Ting Guo

**Affiliations:** Department of Electrical Engineering, National Taiwan University of Science and Technology, Taipei 106335, Taiwan

**Keywords:** temporal multiply-and-sum (TMAS), power Doppler detection, plane-wave (PW) imaging, coherent plane-wave compounding (CPWC), higher-lag autocorrelation, signal coherence, signal decorrelation

## Abstract

Coherent plane wave compounding (CPWC) reconstructs transmit focusing by coherently summing several low-resolution plane-wave (PW) images from different transmit angles to improve its image resolution and quality. The high frame rate of CPWC imaging enables a much larger number of Doppler ensembles such that the Doppler estimation of blood flow becomes more reliable. Due to the unfocused PW transmission, however, one major limitation of the Doppler estimation in CPWC imaging is the relatively low signal-to-noise ratio (SNR). Conventionally, the Doppler power is estimated by a zero-lag autocorrelation which reduces the noise variance, but not the noise level. A higher-lag autocorrelation method such as the first-lag (R(1)) power Doppler image has been developed to take advantage of the signal coherence in the temporal direction for suppressing uncorrelated random noises. In this paper, we propose a novel Temporal Multiply-and-Sum (TMAS) power Doppler detection method to further improve the noise suppression of the higher-lag method by modulating the signal coherence among the temporal correlation pairs in the higher-lag autocorrelation with a tunable *pt* value. Unlike the adaptive beamforming methods which demand for either receive–channel–domain or transmit–domain processing to exploit the spatial coherence of the blood flow signal, the proposed TMAS power Doppler can share the routine beamforming architecture with CPWC imaging. The simulated results show that when it is compared to the original R(1) counterpart, the TMAS power Doppler image with the *pt* value of 2.5 significantly improves the SNR by 8 dB for the cross-view flow velocity within the Nyquist rate. The TMAS power Doppler, however, suffers from the signal decorrelation of the blood flow, and thus, it relies on not only the *pt* value and the flow velocity, but also the flow direction relative to the geometry of acoustic beam. The experimental results in the flow phantom and in vivo dataset also agree with the simulations.

## 1. Introduction

Plane-wave (PW) imaging depends on the unfocused transmit wave to illuminate a wide field of view [1], and the corresponding backscattered echoes are time-compensated for the geometric path of propagation from each receiving channel to the image pixel. When the delayed channel echoes are summed to produce the image output of that pixel, it is referred to as Delay-and-Sum (DAS) beamforming [2]. Since PW imaging with DAS beamforming is intrinsically limited by it having an insufficient image resolution and the rejection of the off-axis clutter due to its unfocused transmit, coherent plane wave compounding (CPWC) has been developed to improve the image quality [3,4]. In CPWC imaging, synthetic transmit focusing is achieved by coherently combining low-resolution images (LRI) to generate the final high-resolution images (HRI). One should note that each LRI is constructed from one distinct PW transmit angle. Assuming that 10 LRI are combined for CPWC at a pulse-repetition-frequency (PRF) of 10 kHz, the resultant HRI in the CPWC imaging will be produced at a frame rate of 1 kHz. Due to this high frame rate, CPWC imaging generally refers to ultrafast imaging, and it has been applied for detecting the motion of imaged objects in transient elastography [4,5] and Doppler flow imaging [6,7,8,9,10,11,12,13]. For example, compared to conventional line-by-line imaging with fixed transmit focusing, the high frame rate of CPWC imaging enables a much larger number of frames to be produced (i.e., Doppler ensembles) during the same amount of observation time such that the estimation of the Doppler parameters becomes more reliable.

Doppler parameters such as power and velocity are routinely used to study the blood flow and they are referred to power Doppler and color Doppler, respectively. Power Doppler provides essential information of the backscattered power of flow signal, and it generally has higher sensitivity to small vessels than color Doppler does [14,15]. Nonetheless, due to the unfocused PW transmission, one major limitation of the Doppler estimation in CPWC imaging is its relatively low signal-to-noise ratio (SNR). Conventionally, the Doppler power in the CPWC image sequence is estimated by the sum-of-square operation. In other words, the power of each frame in the sequence is firstly calculated, and then, incoherent summed across the sequence. This, however, only helps to reduce the noise variance in power Doppler imaging, but not the noise level. Several beamforming methods have been proposed to reduce the noise level of power Doppler imaging in PW imaging by extracting the signal coherence in the spatial direction. For example, the coherent flow power Doppler (CFPD) method [16,17] relies on short-lag spatial coherence [18] to extract the coherence of the blood flow signal in the dimension of the receiving channel, while the Delay Multiply-and-Sum (DMAS) beamforming one is used to highlight the coherence of blood flow signal in the dimension of the PW transmit angles [19,20]. Note that, although the original DMAS beamforming method is implemented by simply multiplying the received echoes between every possible pair [21], alternative high-order versions of DMAS beamforming have been recently proposed to improve the computational efficiency and the flexibility of the tunable image quality [22,23]. For example, when a rational *p* value is used to represent the order of the signal coherence in DMAS beamforming, the efficient DMAS beamforming can be implemented by magnitude scaling of the echoes by *p*-th root and then, taking the *p*-th power of the coherently summed output. In addition to these coherence-based methods, the noise level in power Doppler imaging can also be significantly reduced by the cross-correlation of two complementary subsets of either the receiving channels or the PW transmit angles in the beamforming process [19,24,25,26]. Spatial–temporal denoising of the Doppler image sequence using the non-local mean filter is also proposed to produce a low-noise power Doppler estimation [27]. Though all of the aforementioned methods have been proven to provide effective suppression of the noise level in power Doppler imaging, their implementations generally demand for a more complicated beamforming architecture and/or a more powerful computational capability than that of the conventional CPWC imaging.

Nonetheless, even with the conventional CPWC imaging, an alternative approach to improve the SNR of power Doppler estimation is by exploiting the signal coherence in the temporal direction. According to the knowledge of the authors, the higher-lag autocorrelation method is the first study to investigate the signal coherence in the temporal direction for power Doppler imaging [28]. The higher-lag method relies on the fact that the coherence of flow signal lasts longer over time, while the random noises appear completely uncorrelated even between the adjacent Doppler ensembles. Therefore, instead of using the sum-of-square operation in zero-lag autocorrelation, the magnitude of higher-lag autocorrelation is used to represent the Doppler power. Nonetheless, since a 5-dB decrease in the noise level in the higher-lag method would theoretically require a ten-fold increase in the ensemble number [28], its performance may be limited in the absence of a sufficient number of Doppler ensembles. In this study, a novel Temporal Multiply-and-Sum (TMAS) power Doppler detection is proposed to improve the noise suppression of the higher-lag method. It takes advantage of the concept of DMAS beamforming, but it migrates the estimation of the signal coherence in power Doppler imaging from the spatial direction to the temporal direction. Note that the proposed method can be performed on the conventional CPWC image sequence, and thus, it can be readily incorporated with the existing imaging system without any change in the beamforming process. Section 2 introduces the basics of TMAS power Doppler estimation and its relation to the conventional methods. In Section 3, the simulation methods and experimental setups in this study are described in detail. An achievable improvement to the image SNR of the proposed TMAS power Doppler detection is quantitatively presented in Section 4. Section 5 concludes our results with a discussion.

## 2. Theory

### 2.1. Conventional Power Doppler Imaging Using Zero-Lag Autocorrelation

Ultrasound Doppler detection utilizes repetitive pulse transmissions to observe and to detect the temporal variation of the backscattered blood flow signal. Assume that a total of *M* transmissions are adopted for Doppler detection, the recorded signal from the *m*-th pulse transmission (*m* = 1, 2, …, *M*) is referred to as the *m*-th Doppler ensemble in this study. Note that a clutter filter has to be applied to the Doppler ensembles to separate the blood flow signal from the stationary tissue signal and the thermal noises. In other words, only the signal components with frequencies that are higher than a low-order threshold and lower than a high-order threshold are extracted from the Doppler ensembles for the velocity estimation in color Doppler and/or power estimation in power Doppler. The clutter filter can be a simply temporal filter or a spatial-temporal one such as singular-value decomposition (SVD) [29,30]. Then, the autocorrelation of these clutter-filtered Doppler ensembles is performed for blood flow estimation [31,32] as
(1)$$R(L)=\frac{1}{M}\sum_{m=1}^{M-L}s_{m+L}\;s_{m}^{*}\;\;$$
where $s_{m}$ represents the *m*-th baseband (i.e., IQ) Doppler ensemble after clutter filtering and *L* is the lag number of correlation. For the conventional Doppler power, it is calculated using the zero-lag autocorrelation (R(0)) as
(2)$${\mathrm{PD}}^{L=0}=R(0)=\frac{1}{M}\sum_{m=1}^{M}s_{m}\;s_{m}^{*}=\frac{1}{M}\sum_{m=1}^{M}{\left|s_{m}\right|}^{2}$$

Note that the conventional Doppler power in (2) is simply the normalized incoherent summation of the squared magnitude of all of the *M* Doppler ensembles. Consequently, even with the larger number of Doppler ensembles in PW imaging, the corresponding power Doppler imaging only exhibits a reduced noise variance, but not for the noise level.

### 2.2. Coherence-Based Power Doppler Imaging Using Higher-Lag Autocorrelation

In order to directly reduce the noise level, a coherence-based Doppler power estimation has been proposed [28] by using the magnitude of the higher-lag autocorrelation (i.e., L>0 in (1)) as the Doppler power. By taking the first-lag autocorrelation (R(1)) as an example, the corresponding power Doppler becomes
(3)$${\mathrm{PD}}^{L=1}=\left|R(1)\right|=\left|\frac{1}{M-1\;}\;\sum_{m=1}^{M-1}\;s_{m+1}\;s_{m}^{*}\;\right|$$

Note that (3) actually corresponds to a normalized coherent summation of the complex correlation of neighboring Doppler ensembles. Since the thermal noise de-correlates much faster than the blood flow signal does, the first-lag power Doppler can provide a lower noise background than the conventional zero-lag counterpart can. In other words, since the thermal noise is theoretically independent among each complex correlation due to its random nature, the coherent summation of (3) will lead to a significant cancellation of the noise. This is especially true in PW imaging because the large number of Doppler ensembles helps to ensure the cancellation of random noises.

### 2.3. Higher-Lag Power Doppler Imaging Using Temporal Multiply-and-Sum (TMAS) Method

For the aforementioned higher-lag power Doppler estimation, it has been demonstrated that a zero-mean Gaussian-distributed complex noise will make its mean noise level be inversely proportional to the square root of the ensemble number (i.e., $\sqrt{M}$) [28]. In other words, a ten-fold increase in the ensemble number would lead to a 5-dB decrease in the noise level. Nonetheless, a large number of Doppler ensembles inevitably demand for a longer acquisition time which may not be clinically practical since the velocity and the power of blood flow also change with time. In this study, a temporal multiply-and-sum algorithm is proposed to boost the performance of higher-lag power Doppler imaging without there being excessive numbers of Doppler ensembles. Specifically, instead of the direct summation of the correlations as shown in (3), the multiplication of every possible correlation pair is performed to extract the signal coherence before their summation. Note that, since the correlation pairs themselves are baseband IQ data, the temporal multiply-and-sum algorithm can be performed in a way similar to that of BB-DMAS beamforming [23]. Specifically, the exact implementation of proposed TMAS algorithm begins from the magnitude scaling of the IQ Doppler correlations by *pt*-th root while maintaining their complex phase. Then, the subsequent *pt*-th power is adopted after the coherent summation is performed to include the mutual multiplications among these Doppler correlations. Note that, since the operation of the *pt*-th power will change the dimensionality (i.e., unit) of the Doppler power, this is why the magnitude scaling by *pt*-th root is performed in advance to keep the dimensionality of the Doppler power unchanged. In other words, when the *L*-lag Doppler correlation is represented as $x_{m}^{}=s_{m+L}\;s_{m}^{*}$ (*m* = 1, 2, …, M−L), the *L*-lag power Doppler with TMAS is estimated as
(4)$${\mathrm{PD}}_{\mathrm{TMAS}}^{L}=\left|{\left(\frac{1}{M-L\;}\sum_{m=1}^{M-L}\sqrt[{pt}]{\left|x_{m}\right|}\;e^{j∠x_{m}}\right)}^{pt}\right|$$

In (4), a higher *pt* value in the TMAS algorithm is used to emphasize a more temporal coherence among the Doppler correlations. When *pt* = 1, it should be noted that the *L*-lag power Doppler with the TMAS method in (4) will degenerate to the conventional higher-lag power Doppler as in [28]. In other words, the TMAS power Doppler can be understood as an extension of the conventional higher-lag power Doppler by replacing the simple sum with the multiply and-sum. Note that, in the presence of steady blood flow, the flow component in the Doppler correlations can be modeled as a phasor with a fixed angle, while the noise component is a phasor with a random angle [31]. According to the theory of BB-DMAS [23], the Doppler power with TMAS algorithm (i.e., *pt* > 1) can be approximated by that without TMAS (i.e., *pt* = 1) weighted by a phase coherence factor as demonstrated in the following:(5)$${\mathrm{PD}}_{\mathrm{TMAS}}^{L}={\mathrm{PD}}_{}^{L}*\left|{\left(\frac{1}{M-L\;}\sum_{m=1}^{M-L}\;e^{j∠x_{m}}\right)}^{pt-1}\right|$$

For the flow component, its phase coherence factor will be equal to one because all of the flow phasors are in phase (i.e., $∠x_{m}$= constant for all m). For the noise component, on the contrary, the corresponding phase coherence factor will be less than one because all of the noise phasors are uncorrelated (i.e., $∠x_{m}$ changes randomly among m). Therefore, the TMAS power Doppler will preserve the image pixels in the blood flow region, while those in the background noise region will be relatively suppressed. Note that that the extent of the noise suppression depends on the *pt* value, the number of ensembles and the lag number. Take *pt* = 2 in (4) as an example, the TMAS power Doppler actually comes from the square of the coherent summation of all of the magnitude-scaled Doppler correlations. One should note that, after the square of the summation is multiplied out, there are actually a total of (*M* − *L*)^2^ terms that are coherently combined for the resultant PD estimation. Therefore, it is expected that the case of *pt* = 2 could outperform the counterpart of *pt* = 1 in the suppression of the uncorrelated thermal noise because the number of random noisy components in the coherent summation increases from *M* − *L* to (*M* − *L*)^2^. The signal flowchart of the proposed TMAS power Doppler is schematically represented in Figure 1. Figure 1 shows that the CPWC beamforming is firstly adopted to produce a temporal sequence of HRI. After clutter filtering, the correlations among the temporally neighboring HRI with a lag number of *L* are computed. Here, the first-lag correlation is demonstrated in Figure 1 as an example, but it can be replaced by any other higher-lag correlation according to the flow velocity that is of interest. Finally, the multiply-and-sum operation is performed on these correlations to highlight the temporal coherence in the power Doppler detection. It should be emphasized that due to the baseband nature of Doppler signal, the TMAS signal processing in Figure 1 does not generate a DC or harmonic component as in [21] and it does not demand for any filters.

## 3. Methods

### 3.1. Simulation Setup

In this study, the blood flow data are simulated using Field II simulation program [33,34] with the parameters that are shown in Table 1. A flow vessel in the speckle-generating tissue phantom is constructed with a radius of 2 mm and an inclined angle (θ) of 30° and 0° to simulate the blood flow with and without the axial component, respectively. The flow vessel is positioned to be relative to the array transducer to provide the cross-view and the longitudinal-view images, respectively, as depicted in Figure 2a,b. The laminar flow in the vessel has peak velocities of 150, 310 and 700 mm/s, and the scattering magnitude of the flow is assumed to be 60 dB lower than that of the surrounding stationary tissue. At the inclined angle of 30°, it should be noted that the axial flow velocity in the case of 150 mm/s is still within the Nyquist rate of Doppler detection in this study. For the cases of 310 and 700 mm/s, on the contrary, aliasing artifacts would occur in the color Doppler imaging. In this study, since only the power Doppler imaging is considered, the aliasing of the flow signal is not a problem here, and these high-velocity flow conditions are purposely included to investigate the decorrelation of the flow signal in TMAS power Doppler detection. White Gaussian noises are also added into the channel data before dynamic they receive beamforming to produce a channel signal-to-noise of 0 dB for the flow signal. The channel data are received with a 128-element linear array transducer (pitch = 0.3 mm and center frequency = 5 MHz). A total of 6 plane waves (−5° to +5°) are sequentially transmitted with a PRF of 6 kHz to produce LRI from distinct PW angles. The HRI are then obtained by coherent compounding LRI as in CPWC to achieve a frame rate of 1 kHz. A total of 64 frames (i.e., Doppler ensembles) are acquired. For HRI, the SVD filter is used to remove the stationary tissue and noise signals. In simulation, the low-order and high-order thresholds of the SVD clutter filter are, respectively, set to be 2 and 30. Finally, the power Doppler image was calculated using the proposed TMAS method in this study.

### 3.2. Experimental Setup

In the phantom experiment, the imaging parameters are the same as those in the simulation setup. The commercial flow phantom is mini-Doppler 1430 (Sun Nuclear, Melbourne, FL, USA), which provides one horizontal tube and the other one which is inclined by 35°. Two constant velocities of 310 and 700 mm/s are considered in the flow phantom. Note that the flow velocity of 310 mm/s is close to the lowest achievable velocity in the flow phantom. The linear array has a pitch of 0.3 mm, and it was excited by a 5-cycle sinusoidal burst at 5 MHz. For the phantom experiment, five independent measurements were performed for statistical analysis. In vivo rabbit kidney data from 6-month-old female New Zealand white rabbits were provided by the S-Sharp Corporation (New Taipei, Taiwan). The rabbit was anesthetized by intramuscular injecting Zoletil^®^ 50 and its body temperature was maintained at 37 °C with a warming pad during the experiments. The animal study was approved by the Institutional Animal Care and Use Committee of National Taiwan University, Taipei, Taiwan. The in vivo dataset was acquired with the same PW sequence, but at a PRF of 4 kHz. Therefore, the frame rate is approximately 667 Hz. Nonetheless, a higher transmit frequency (i.e., 6.4 MHz) was used in the in vivo dataset to provide sufficient the spatial resolution of smaller vessels in the rabbit kidney. Note that, for both the phantom and in vivo experiments, the LRI were constructed and recorded using Prodigy ultrasonic imaging system (S-sharp, New Taipei City, Taiwan) for off-line coherent compounding. Then, the compounded HRI are SVD were clutter filtered with the low-order and high-order thresholds of (6, 30) and (10, 40), respectively, in the phantom and in vivo experiments. MATLAB software (The MathWorks, Natick, MA, USA) was used to process the data from the SVD filtering to the power Doppler estimation. The detailed experimental imaging parameters are shown in Table 2.

### 3.3. Quantitative Analysis

To quantify the effect of the *pt* value and the lag number in the proposed TMAS power Doppler imaging, two respective regions of interest in the blood flow area and the background noise area are used to estimate the image SNR as defined as follows:$$SNR=10\;\log_{10}\left(\frac{{\bar{M}}_{blood}}{{\bar{M}}_{noise}}\right)\;$$
where ${\bar{M}}_{blood}$ and ${\bar{M}}_{noise}$ are the mean power of blood flow signal and background noise, respectively, which are indicated by the blue and the white boxes in the power Doppler images of this study.

## 4. Results

### 4.1. Simulations

Cross-view TMAS power Doppler images of the simulated flow phantom with the lag numbers of 0, 1 and 2 are provided in the upper, middle and lower panels of Figure 3 and are denoted as R(0), R(1) and R(2), respectively. The vessel is inclined by 30° relative to the horizon, and the flow velocity in the simulation is 150 mm/s. The power Doppler images from left to right correspond to different *pt* values of 1.0, 1.5, 2.0 and 2.5 for all of the lag numbers of correlation. It should be noted that the images with a *pt* value of 1.0 are provided as a reference to the proposed TMAS method. For the R(0) images, the case with a *pt* value of 1.0 is the conventional power Doppler as in (2), while it is the original higher-lag power Doppler images as in (3) for R(1) and R(2) images. All of the images were normalized to the maximum of the conventional power Doppler image (i.e., the uppermost and the leftmost panel) to demonstrate their relative change in the Doppler power. The display dynamic range is fixed to 40 dB for all of the power Doppler images in this study. One should note that the Doppler power in the background region (i.e., the white box) is contributed by only noises since it does not enclose any blood flow in the vessel.

The results in Figure 3 indicate that the noise level in the background region of the higher-lag images such as R(1) and R(2), which consistently decrease with the *pt* value. For R(1), this leads to the increase in the SNR from 22.0 dB in reference to 24.8 dB, 27.5 dB and 30.0 dB in the TMAS method, respectively, with the *pt* values of 1.5, 2.0 and 2.5. For R(2), the SNR increases from 21.4 dB in reference to 24.0 dB, 26.3 dB and 28.4 dB in the TMAS method, respectively, with the *pt* values of 1.5, 2.0 and 2.5. It should be noted that with the same *pt* value, the SNR of the R(2) image is consistently lower than that of the R(1) counterpart. This is because the R(2) image suffers from relatively higher signal decorrelation, and thus, it has a lower Doppler power in the blood flow region. For example, when the *pt* value is 2.5, it is visually detectable that the brightness in the blood flow region decreases in the R(2) image relative to the R(1) image. Nonetheless, for the R(0) images, it is also noticeable that both the signal and the noise levels appear to be unchanged with the *pt* value, and thus, the corresponding SNR remains virtually the same.

We further increased the flow velocity in the simulation to 310 mm/s to investigate the performance of TMAS power Doppler for high-velocity flow in Figure 4. Compared to the low-velocity counterpart in Figure 3, the TMAS power Doppler images in Figure 4 show more evident signal decorrelation especially for the higher-lag images. It is noticeable that because of the laminar distribution of the flow velocity, the signal decorrelation in the center of the vessel is much higher than it is that near the vessel wall, and thus, this leads to the donut-shaped brightness distribution in the TMAS power Doppler images. This phenomenon becomes more obvious with both the lag number and the *pt* value. Consequently, even though the SNR still increases with the *pt* value for high-velocity flow in Figure 4, the achievable SNR improvement became marginal. For example, when the *pt* value increased from 1.0 to 2.5, the SNR of R(1) image improved by 8 dB at the flow velocity of 150 mm/s, as shown in Figure 3, but the corresponding SNR improvement at the flow velocity of 310 mm/s reduces to 6.8 dB. Since the R(2) images suffer from more severe decorrelation in the blood flow region, the loss of the SNR improvement becomes even more evident. Specifically, when the *pt* value increased from 1.0 to 2.5, the SNR of R(2) image improved by 7 dB at the flow velocity of 150 mm/s, as in Figure 3, while the corresponding SNR improvement reduced to only 1.8 dB at the flow velocity of 310 mm/s.

In order to separate the effect of signal decorrelation from that of noise suppression in the TMAS power Doppler images, the signal intensities of the blood flow region in Figure 3 and Figure 4 are provided as functions of *pt* values in Figure 5 for both of the flow velocities of 150 mm/s and 310 mm/s. Note that each curve is normalized to the value with the *pt* value of 1.0 for us to better compare the extent of the signal decorrelation at different flow velocities. As expected, Figure 5 demonstrates that the R(1) signal intensity in the cross-view decreases with the *pt* value at both of the flow velocities, and the decrease at 310 mm/s is consistently higher than that at 150 mm/s. The comparison between the R(1) and the R(2) signal intensity also confirms that at the same flow velocity, there is more severe signal decorrelation in the R(2) images than there is in the R(1) images. For example, when the *pt* value increases from 1.0 to 2.5 at the flow velocity of 310 mm/s, the signal decorrelation in the cross-view TMAS power Doppler leads to a signal loss of about 6 dB in the R(1) images, while the signal loss escalates to about 12 dB in the R(2) images. For the noise level, on the contrary, it should be noted that the noise background of the R(1) images in both the Figure 3 and Figure 4 remains similar to that of the R(2) images for the same *pt* value. Since the R(1) images generally suffer less signal decorrelation but they still provide similar noise suppression to the R(2) counterparts, only the R(1) images will be considered and presented in the following of this paper.

To simplify the simulation for the noises in the higher-lag power Doppler estimation, an image sequence comprising of only white noise was directly generated using the MATLAB awgn function to validate the achievable noise suppression as a function of the ensemble number. The size of each image in the sequence is 150 by 150 in pixels, and the length of the sequence is 1000 ensembles. The noise level is estimated by the spatial mean of the entire higher-lag power image without or with the TMAS algorithm. Here, only the first-lag power (i.e., R(1)) is considered. One should note that the noise level with the *pt* value of 1.0 corresponds to the original higher-lag power estimation without the TMAS algorithm. As expected in [28], Figure 6 shows that the R(1) noise level without TMAS does decrease by 5 dB when the ensemble number increases by ten-fold. When the temporal coherence is further emphasized by including the TMAS algorithm in the higher-lag power Doppler, the achievable noise suppression increases with the *pt* value. This can be noted by the faster decay of the noise level with TMAS than that which occurs without TMAS. Specifically, the decay of the noise level with TMAS with a ten-fold increase of the ensemble number is 7.7, 10.2 and 12.8 dB, respectively, for the *pt* values of 1.5, 2.0 and 2.5. This demonstrates that the proposed TMAS algorithm can help to effectively suppress more uncorrelated noise through the flexible selection of the *pt* value. Moreover, it should be noted that the decay of noise level for *p**t* = 2.0 roughly doubles that for *p**t* = 1.0. This is consistent with our expectation when the number of random noisy components in the coherent summation increases from *M* − 1 to (*M* − 1)*^2^* by using the TMAS algorithm with *p**t* = 2.0.

In order to further examine how the signal decorrelation changes with the flow orientation, the longitudinal-view TMAS power Doppler images of the simulated 30° flow phantom are also provided in Figure 7 at both of the flow velocities of 150 mm/s and 310 mm/s. The conventional power Doppler image (i.e., R(0) with the *pt* value of 1) is also demonstrated in the leftmost panel for a comparison to be made. As they are similar to those in cross view, the results in longitudinal view also show that the SNR of the R(1) images markedly increases with the *pt* value at both of the flow velocities. At the flow velocity of 150 mm/s, the Doppler power within the vessel remains visually uniform which indicates that the signal decorrelation is not evident even when a high *pt* value (e.g., 2.5) is adopted. At the flow velocity of 310 mm/s, on the contrary, a noticeable signal decorrelation occurs in the center line of the vessel and gradually leads to a hollow appearance of the vessel when the *pt* value increases. The longitudinal-view R(1) and R(2) signal intensities of the blood flow region at the flow velocity of 310 mm/s are also included in Figure 5. It is shown that, when the vessel is inclined by 30° to introduce the axial component of blood flow, the signal intensity of the longitudinal view is barely distinguishable from that of cross view for both the R(1) and R(2) images. One should note that the non-axial flow in the cross view is in the elevational direction, while that in the longitudinal view is in the lateral direction. In other words, in the presence of axial flow, the effect of the non-axial flow direction on signal decorrelation is relatively minor. Moreover, it should be noted that the longitudinal-view TMAS power Doppler images in Figure 7 also exhibit more apparent signal decorrelation in the left end of the vessel than they do in the right end. This is because the angle to the center point of the receiving array is smaller in the left end (i.e., $\phi_{1}\;<\;\phi_{2}$ as drawn in Figure 7), and thus, the axial velocity of the flow that is perceived by the receiving array should be faster in the left than it is in the right. This observation further confirms that the axial velocity is a more dominant factor for signal decorrelation than the non-axial velocity is.

Additional simulations were also performed for a horizontal flow vessel (i.e., 0° vessel) to study the effect of non-axial flow direction on signal decorrelation in TMAS power Doppler imaging without any axial component. Figure 8 and Figure 9 demonstrates the corresponding TMAS power Doppler images at both the flow velocities of 310 mm/s and 700 mm/s, respectively, in the cross and the longitudinal views. Compared to the counterparts of 30° vessel, the R(1) images of 0° vessel in the upper panels of Figure 8 and Figure 9 show negligible signal decorrelation even at the high flow velocity of 310 mm/s. This observation confirms that the axial flow is indeed the dominant factor for signal decorrelation in the TMAS power Doppler images. When the flow velocity is further increased to 700 mm/s, which is shown in the lower panels of Figure 8 and Figure 9, to purposely exaggerate the signal decorrelation with only the non-axial flow, the longitudinal-view TMAS power Doppler images begin to visually demonstrate noticeable signal decorrelation, while the cross-view counterparts do not.

The corresponding signal intensity of the blood flow region in Figure 8 and Figure 9 is also demonstrated in Figure 10 as a function of the *pt* value for the 0° vessel. The results in Figure 10 confirm a more severe signal decorrelation in the longitudinal view than they do in the cross view. For example, when the *pt* value increases from 1 to 2.5, the loss of the R(1) signal intensity at the flow velocity of 700 mm/s is about 5 dB in the longitudinal view, while it is about only 2 dB in the cross view. Similarly, the corresponding loss of the R(2) signal intensity is about 8 dB in the longitudinal view, while it is less than 3 dB in the cross view. This observation suggests that the blood flow in the longitudinal view evidently introduces more signal decorrelation than it does in the cross view in the TMAS power Doppler images. In other words, in the absence of axial flow, the direction of non-axial flow also has impacts on the degree of signal decorrelation. It is reasonable to observe that the lateral flow leads to higher signal decorrelation than the elevational flow does because the beam width in the lateral direction is generally narrower than it is in the elevational direction due to both the dynamic receive beamforming and the coherent compounding of multiple plane wave transmissions. Consequently, due to the low signal decorrelation of the 0° vessel in the cross view, the difference in the cross-view signal intensity between the flow velocities of 310 mm/s and 700 mm/s is negligible (i.e., <1 dB) in Figure 10 for both the R(1) and R(2) images.

### 4.2. Experiments

The TMAS power Doppler images of the horizontal 0° vessel in the commercial flow phantom are provided in Figure 11 and Figure 12, respectively, for the cross view and longitudinal view at both flow velocities of 310 and 700 mm/s. Similar to the corresponding simulation results in Figure 8 and Figure 9, the power Doppler images of the experimental phantom also show more severe signal decorrelation in the longitudinal view than they do in the cross view. This can be visually identified by the rapid decrease of the image brightness in the blood flow region with the pt value in the longitudinal view. The signal decorrelation also increases with the flow velocity, as was expected in the simulations. This can be visually identified by the rapid decrease in the image brightness in the blood flow region with the pt value in the longitudinal view. The signal decorrelation also increases with the flow velocity, as was expected in the simulations. This is particularly noticeable for the longitudinal-view power Doppler images in Figure 12 which demonstrate a significant loss of the signal intensity not only in the center line, but within the entire vessel. This suggests that the blood flow in the experimental vessel may not exhibit the ideal laminar distribution as it does in the simulation. Consequently, even though the R(0) reference image at the flow velocity of 310 mm/s has a similar SNR between the cross view (12.7 dB) and the longitudinal view (11.5 dB), the SNR of cross-view R(1) image can significantly improve to 25.9 dB with the *pt* value of 2.5, while the longitudinal-view counterpart only improves to 18.4 dB.

The TMAS power Doppler images in the presence of axial flow are demonstrated using the dataset of the rabbit’s kidney in Figure 13. Visual observations also show that the power Doppler images with TMAS generally have a lower noise level in the background than those that were obtained with the conventional methods. On the contrary, the decorrelation of the blood signal due to the flow in the axial direction is less noticeable because the estimated flow velocity remains within the Nyquist rate in the dataset of the rabbit’s kidney. Consequently, the experimental SNR increases from 21.0 dB in R(0) image to 22.7 dB, 23.6 dB, 24.5 dB and in 25.5 dB, respectively, with the *pt* values of 1.0, 1.5, 2.0 and 2.5. Taking the *pt* value of 2.5 as an example, the improvement in the Doppler SNR is 4.5 dB when we are compared it to the reference R(0) image. Nonetheless, it should be noted that the R(1) image with the *pt* value of 2.5 also provides an SNR improvement of 2.8 dB when we are compared it to the conventional higher-lag method (i.e., R(1) image with the *pt* value of 1.0).

## 5. Discussion and Conclusions

In this study, a novel power Doppler imaging methos using the TMAS approach has been investigated using both simulations and experiments. Unlike the conventional higher-lag counterpart which uses a simple summation of all of the correlation pairs for the power Doppler estimation, the proposed TMAS power Doppler imaging relies on the mutual multiplication of these correlation pairs to further extract the signal coherence. One should note that it is the correlation pairs, not the Doppler ensembles, are used for mutual multiplications in the TMAS power Doppler. This is because the direct multiplication of the Doppler ensembles will suffer from the motion interference of that of the blood flow signal. On the contrary, since each correlation pair in the R(1) image is estimated by the product of the (*m* + 1)-th ensemble with the conjugate of the m-th ensemble, its phase actually comes from the phase difference between the two consecutive ensembles. This makes the blood flow signal in the correlation pairs have a higher temporal coherence than they do in the original Doppler ensembles. For example, if we are assuming that there is a constant flow velocity over time, all of the correlation pairs theoretically should have the same phase term and become highly coherent in the temporal domain. One should recall that it is exactly the phase term in the first-lag correlation pairs which determines the estimated flow velocity in auto-correlation methods [31]. Though the number of Doppler ensembles and the corresponding correlation pairs which can be large in the ultrafast PW imaging, the implementation of the TMAS algorithm can be efficiently performed by the magnitude scaling of the Doppler correlations by *pt*-th root and the subsequent *pt*-th power after a coherent summation to restore the dimensionality of the Doppler power. One should note that the proposed TMAS algorithm itself does not demand for any channel–domain or transmit–domain processing to occur, and thus, it can share the routine DAS beamforming with the CPWC imaging. In other words, the TMAS power Doppler is fully compatible with the current CPWC imaging system and can be regarded as an extension of the conventional higher-lag power Doppler.

The results indicate that, when the *pt* value increases to emphasize the higher the signal coherence of the Doppler ensembles in the temporal domain, the TMAS power Doppler is capable of effective SNR improvement for higher-lag images such as R(1) and R(2). The same benefit, however, is not observable in a zero-lag R(0) image. This is because each frame of the R(0) image has become a power image itself with only a positive pixel value. Since any positive value has the same phase of zero, the coherence of the noises among the temporal frames of R(0) now becomes elevated, and thus, the TMAS algorithm cannot separate the noises from the flow signal using their difference in temporal coherence. Specifically, since the noises in each frame of the R(0) image no longer have a zero-mean distribution, any denoising approach can only reduce the noise variance, but not the noise level. The same observation has also been reported in the attempt to apply the non-local-mean filtering on the R(0) image [27].

Nonetheless, our results also suggest that the decorrelation of the blood flow signal may compromise the achievable SNR improvement in TMAS power Doppler imaging. When only the non-axial flow is considered, the lateral flow leads to more signal decorrelation than the elevational flow does because the acoustic beam is narrower in the lateral direction than it is in the elevational direction. In the presence of the axial flow component, the decorrelation of the blood flow could become even more severe especially at excessive flow velocities. In order to avoid too much signal decorrelation, it is suggested that the flow velocity of the axial component should be within the Nyquist rate. In our simulation setup, since the inclined angle of the vessel is 30°, the flow velocity of 150 mm/s actually corresponds to the Nyquist rate of 75 mm/s in this study. In this case, Figure 5 shows that the loss of the R(1) signal intensity due to decorrelation is about only 3 dB even with the *pt* value of 2.5. Nonetheless, the corresponding loss of the R(2) signal intensity deteriorates for it to be about 6 dB. Since the noise reduction in the R(1) images remains comparable to that in the R(2) images, it is also suggested that researchers should adopt R(1) images in the TMAS power Doppler estimation to better maintain the signal intensity of the blood flow. One should note that the R(1) images in the TMAS power Doppler estimation actually corresponds to the short-range coherence imaging as suggested in [28] for blood flow with a high-flow velocity. For the capillary perfusion, TMAS power Doppler imaging could also outperform the conventional higher-lag imaging method. Compared to the conventional higher-lag image without TMAS, the high-velocity blood disappears faster in the TMAS image, and thus, the capillary perfusion can be separated from the conduit flow even with a lower lag number. One should note that, for the lag number of *L* and the total ensemble number of *M*, the available number of correlation pairs is (*M* − *L*). With a lower *L* in the TMAS power Doppler estimation, the achievable suppression of the noise level would further increase in the perfusion imaging due to the increased number of correlation pairs.

The proposed TMAS power Doppler estimation may be applicable to not only DAS beamforming as demonstrated in this work, but also other adaptive beamforming methods. For example, when the TMAS power Doppler estimation is combined with any beamforming, which exploits the spatial coherence of blood flow signal in either the dimension of the receiving channel or the PW transmit angles such as in [16,17,19,20,24,25,26], the corresponding power Doppler imaging would rely on the spatial–temporal coherence of the blood flow signal to produce the image pixel. In this case, a smaller *pt* value may be used in a TMAS algorithm to alleviate the signal decorrelation of high-velocity blood flow, while the overall noise suppression remains unchanged due to the inclusion of spatial coherence in the beamforming stage. This spatial–temporal coherence of the power Doppler estimation will be the focus of our future work.

## Figures and Tables

**Figure 1 sensors-22-08349-f001:**
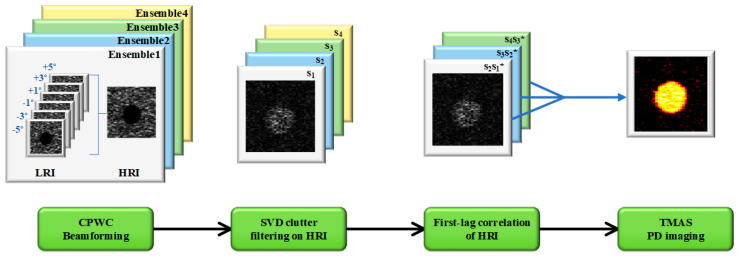
Schematic diagram of the proposed TMAS power Doppler detection in multi-angle PW imaging. Note that the correlation can be changed from first-lag to other higher-lag according to the flow velocity of interest. The symbol * represents complex conjugate.

**Figure 2 sensors-22-08349-f002:**
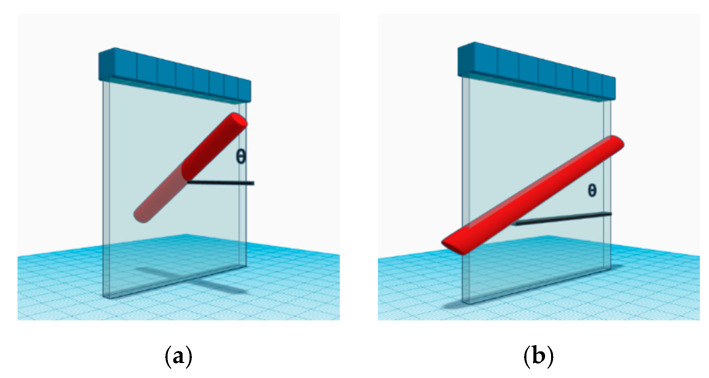
Schematic diagram of the simulated flow vessel in the speckle phantom with an inclined angle of θ for (**a**) cross-view imaging and (**b**) longitudinal-view imaging.

**Figure 3 sensors-22-08349-f003:**
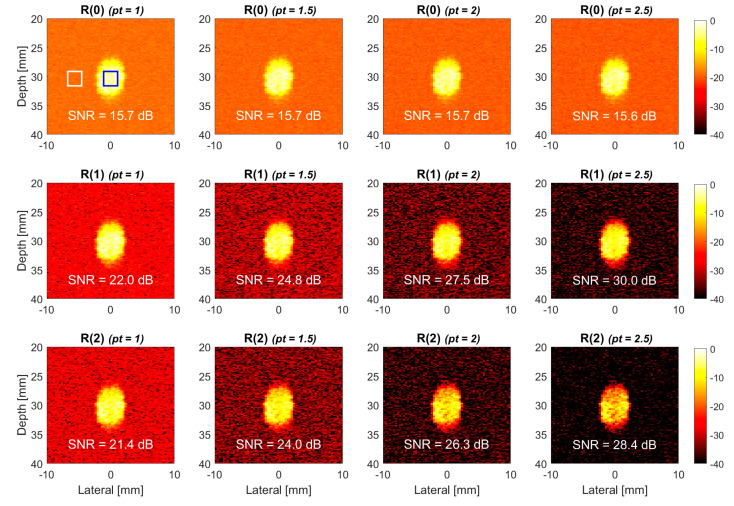
Simulated TMAS power Doppler images of cross-view flow phantom for R(0), R(1) and R(2) images with the *pt* values of 1.0, 1.5, 2.0 and 2.5, respectively, from left to right. The flow velocity is 150 mm/s, and the inclined angle is 30°.

**Figure 4 sensors-22-08349-f004:**
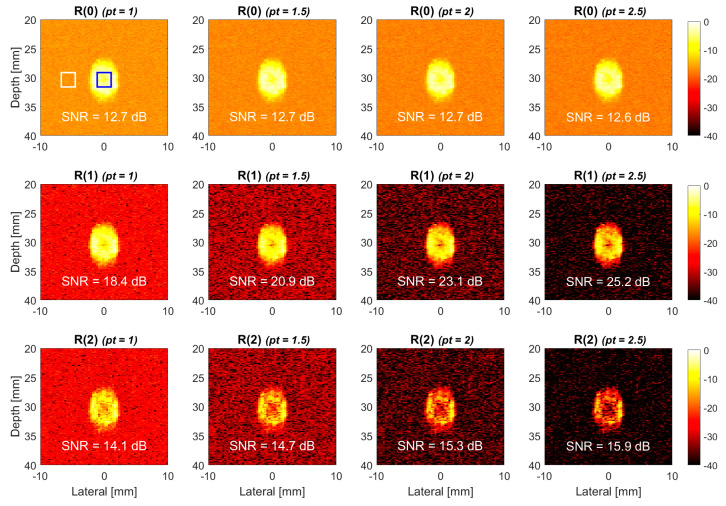
Simulated TMAS power Doppler images of cross-view flow phantom for R(0), R(1) and R(2) images with the *pt* values of 1.0, 1.5, 2.0 and 2.5, respectively, from left to right. The flow velocity is 310 mm/s, and the inclined angle is 30°.

**Figure 5 sensors-22-08349-f005:**
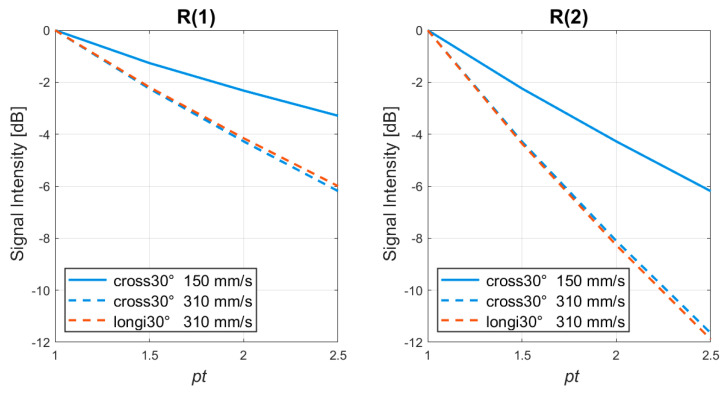
Simulated R(1) and R(2) signal intensity of the blood flow as a function of the *pt* value in TMAS power Doppler detection. The inclined angle of the flow vessel is 30°.

**Figure 6 sensors-22-08349-f006:**
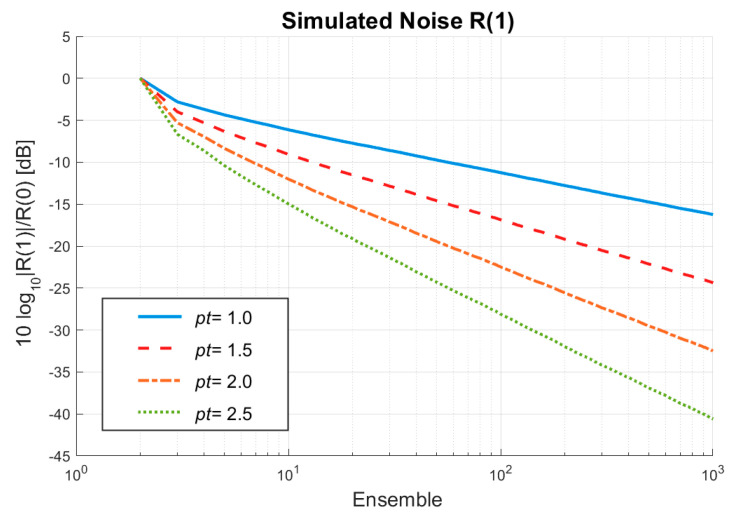
Simulated noise level as a function of the ensemble number for the *pt* values of 1.0, 1.5, 2.0 and 2.5 in TMAS power Doppler detection.

**Figure 7 sensors-22-08349-f007:**
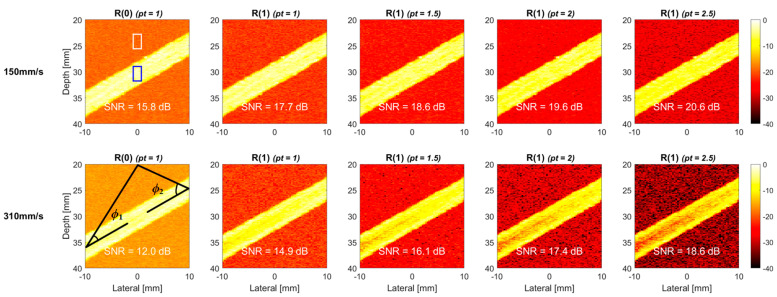
Simulated TMAS power Doppler images of longitudinal-view flow phantom for R(1) images with the *pt* values of 1.0, 1.5, 2.0 and 2.5, respectively, from left to right. R(0) image is also provided as a reference. The flow velocity is 150 mm/s (**upper**) and 310 mm/s (**lower**). The inclined angle of the flow vessel is 30°.

**Figure 8 sensors-22-08349-f008:**
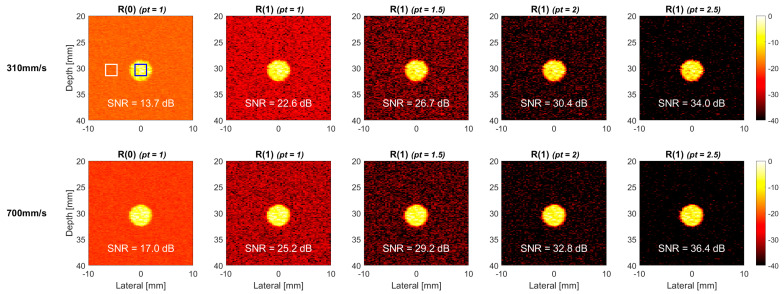
Simulated TMAS power Doppler images of cross-view flow phantom for R(1) images with the *pt* values of 1.0, 1.5, 2.0 and 2.5, respectively from left to right. R(0) image is also provided as a reference. The flow velocity is 310 mm/s (**upper**) and 700 mm/s (**lower**). The inclined angle of the flow vessel is 0°.

**Figure 9 sensors-22-08349-f009:**
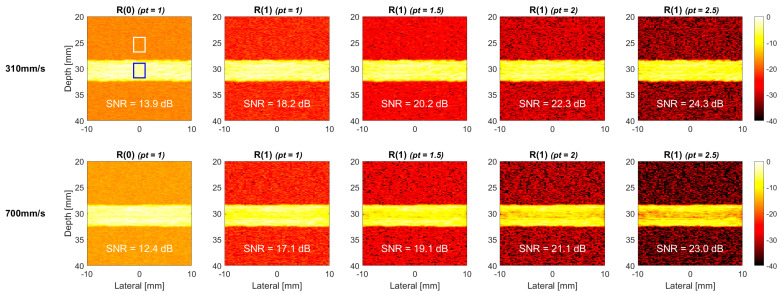
Simulated TMAS power Doppler images of longitudinal-view flow phantom for R(1) images with the *pt* values of 1.0, 1.5, 2.0 and 2.5, respectively from left to right. R(0) image is also provided as a reference. The flow velocity is 310 mm/s (**upper**) and 700 mm/s (**lower**). The inclined angle of the flow vessel is 0°.

**Figure 10 sensors-22-08349-f010:**
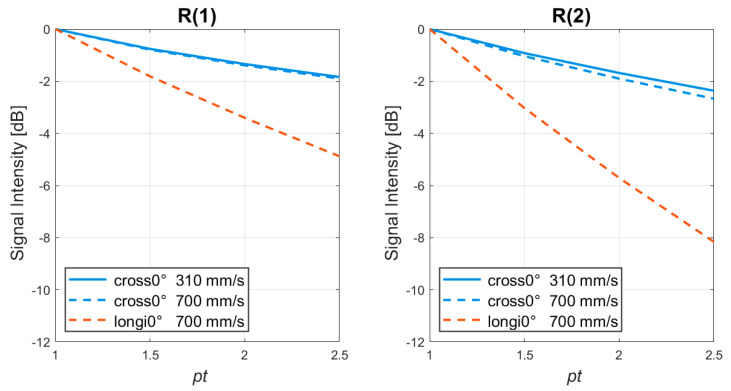
Simulated R(1) and R(2) signal intensity of the blood flow as a function of the *pt* value in TMAS power Doppler detection. The inclined angle of the flow vessel is 0°.

**Figure 11 sensors-22-08349-f011:**
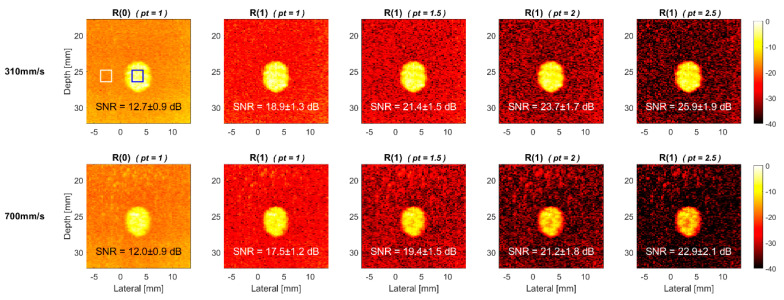
Experimental TMAS power Doppler images of cross-view flow phantom for R(1) images with the *pt* values of 1.0, 1.5, 2.0 and 2.5, respectively from left to right. R(0) image is also provided as a reference. The flow velocity is 310 mm/s (**upper**) and 700 mm/s (**lower**). The inclined angle of the flow vessel is 0°.

**Figure 12 sensors-22-08349-f012:**
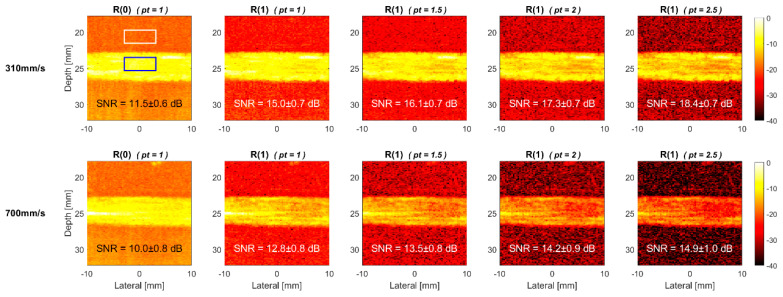
Experimental TMAS power Doppler images of longitudinal-view flow phantom for R(1) images with the *pt* values of 1.0, 1.5, 2.0 and 2.5, respectively from left to right. R(0) image is also provided as a reference. The flow velocity is 310 mm/s (**upper**) and 700 mm/s (**lower**). The inclined angle of the flow vessel is 0°.

**Figure 13 sensors-22-08349-f013:**
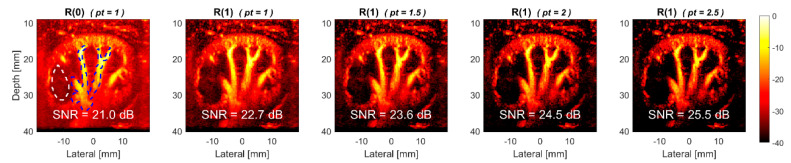
Experimental TMAS power Doppler images of in vivo rabbit’s kidney for R(1) images with the *pt* values of 1.0, 1.5, 2.0 and 2.5, respectively from left to right. R(0) image is also provided as a reference.

**Table 1 sensors-22-08349-t001:** Imaging parameters in FIELD II simulations.

Imaging System	
Transducer	Linear Array
Pitch	0.3 mm
Number of elements	128
Elevation focus	30 mm
Sampling frequency	20 MHz
Image size in pixels	275 (axial) × 128 (lateral)
**Transmit Pulse**	
Center frequency	5.0 MHz
Excitation	3 cycles
PW transmit angle	6 (−5°~+5°)
Ensemble	64
PRF	6.0 kHz
**Phantom**	
Speed of Sound	1550 m/s
Scattering magnitude	60 dB (tissue clutter)
	0 dB (blood flow)

**Table 2 sensors-22-08349-t002:** Imaging parameters in phantom and in vivo experiment.

Prodigy Imaging System	
Transducer	L154BH
Pitch	0.3 mm
Number of elements	128
Elevation focus	20 mm
Sampling frequency	25.6 MHz
Image size in pixels	520 (axial) × 128 (lateral)
**Transmit Pulse**	
Center frequency	5 MHz (phantom)
	6.4 MHz (in vivo)
Excitation	5 cycles
PW transmit angle	6 (−5°~+5°)
Ensemble	64
PRF	6 kHz (phantom) 4 kHz (in vivo)

## Data Availability

Restrictions apply to the availability of these data. Experimental data were obtained from S-Sharp Corporation (New Taipei, Taiwan) and are available with the permission of S-Sharp Corporation.

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
