# Peer review of "Ultrasound Ultrafast Power Doppler Imaging with High Signal-to-Noise Ratio by Temporal Multiply-and-Sum (TMAS) Autocorrelation"

_sensors, 2022, doi:10.3390/s22218349_

Round 1
Reviewer 1 Report
Please find the attached comments.

Author Response
The authors appreciate the comments from reviewer. Please see the attached file for our detailed responses to these comments.

Reviewer 2 Report
In this manuscript, the authors proposed a temporal multiply-and-sum (TMAS) method to improve the signal-to-noise (SNR) ratio in ultrasound ultrafast power Doppler images. The results, especially the simulations, indicate that the proposed method can significantly improve the SNR. However, the novelty of this manuscript was not very well demonstrated. The authors used only a little more than half page in Section 2.3 to explain their proposed method. Equation (4), which is critical in this study, does not seem quite right. Section 2.1 and 2.2 are more like common basis of Doppler imaging. It is suggested that the authors describe their proposed method in detail or use more mathematical equations to explain why the method is able to improve the SNR.
Author Response

(The authors gave the same response as above.)

Round 2
Reviewer 2 Report
The authors have made revisions according to the reviewer's comments. However, I still encourage the authors to further strengthen the description of the proposed method. For example, what is the characteristics of the signal? Why would magnitude scaling help reduce noise level? What is the theoretical foundation of this method?
